# A biofilm-tropic *Pseudomonas aeruginosa* bacteriophage uses the exopolysaccharide Psl as receptor

Brenna Walton[1], Serena Abbodante[2,3], Michaela Ellen Marshall[2,3], Justyna M Dobruchowska[4], Amani Alvi[1], Larry A Gallagher[5], Nikhil Vallikat[1], Zhemin Zhang[6], Daniel J Wozniak[7,8], Edward W Yu[6], Geert-Jan Boons[4,9,10], Eric Pearlman[2,3], Arne Rietsch[1]*

[1]Department of Molecular Biology and Microbiology, Case Western Reserve University, Cleveland, United States; [2]Department of Ophthalmology, University of California, Irvine, Irvine, United States; [3]Ophthalmology and Visual Sciences, Institute for Immunology, University of California, Irvine, Irvine, United States; [4]Department of Chemical Biology and Drug Discovery, Utrecht Institute for Pharmaceutical Sciences, and Bijvoet Center for Biomolecular Research, Utrecht University, Utrecht, Netherlands; [5]Department of Microbiology, University of Washington, Seattle, United States; [6]Department of Pharmacology, Case Western Reserve University, Cleveland, United States; [7]Department of Microbial Infection and Immunity, The Ohio State University, Columbus, United States; [8]Department of Microbiology, The Ohio State University, Columbus, United States; [9]Complex Carbohydrate Center, University of Georgia, Athens, United States; [10]Department of Chemistry, University of Georgia, Athens, United States

*For correspondence:
arne.rietsch@case.edu

Competing interest: The authors declare that no competing interests exist.

## eLife Assessment

This **valuable** study identifies a novel bacteriophage that can use the exopolysaccharide Psl of *Pseudomonas aeruginosa* to infect and disrupt biofilms. The work is **convincing** and suggests a novel approach to control biofilms that is relevant to researchers working on biofilms, specifically in Pseudomonas, on phage physiology and discovery, and on alternatives to controlling bacterial pathogens.

**Abstract** Bacteria in nature can exist in multicellular communities called biofilms. Biofilms also form in the course of many infections. *Pseudomonas aeruginosa* infections frequently involve biofilms, which contribute materially to the difficulty to treat these infections with antibiotic therapy. Many biofilm-related characteristics are controlled by the second messenger, cyclic-di-GMP, which is upregulated on surface contact. Among these factors is the exopolysaccharide Psl, which is a critically important component of the biofilm matrix. Here, we describe the discovery of a *P. aeruginosa* bacteriophage, which we have called Clew-1, that directly binds to and uses Psl as a receptor. While this phage does not efficiently infect planktonically growing bacteria, it can disrupt *P. aeruginosa* biofilms and replicate in biofilm bacteria. We further demonstrate that the Clew-1 can reduce the bacterial burden in a mouse model of *P. aeruginosa* keratitis, which is characterized by the formation of a biofilm on the cornea. Due to its reliance on Psl for infection, Clew-1 does not actually form plaques on wild-type bacteria under standard in vitro conditions. This argues that our standard isolation procedures likely exclude bacteriophage that are adapted to using biofilm markers for infection. Importantly, the manner in which we isolated Clew-1 can be easily extended to other strains of *P.*

*aeruginosa* and indeed other bacterial species, which will fuel the discovery of other biofilm-tropic bacteriophage and expand their therapeutic use.

## Introduction

Biofilms formed by bacteria at sites of infection significantly increase the difficulty of treatment with conventional antibiotic therapy. This increased resistance to antibiotic therapy has been attributed to a variety of factors, including reduced penetration of antibiotics (*Wilton et al., 2016*; *Chiang et al., 2013*), as well as an increase in antibiotic-tolerant persister bacteria (*Ciofu et al., 2022*; *Lewis, 2008*). Formation of biofilms is a feature of many *P. aeruginosa* infections, including lung infections in cystic fibrosis patients (*Høiby et al., 2010*; *Mulcahy et al., 2010*), wound, catheter, and device infections (*Ledizet et al., 2012*), as well as blinding corneal infections (*Thanabalasuriar et al., 2019*; *Behlau and Gilmore, 2008*; *Okurowska et al., 2024*). In some instances, these biofilms have been found to be astonishingly antibiotic-tolerant (*Hamed and Debonnett, 2017*).

In addition to the antibiotic tolerance of bacteria in biofilms, there has been a significant increase in antibiotic-resistant isolates (*D'Agata, 2004*). In fact, *P. aeruginosa* is one of the particularly worrisome ESKAPE group of pathogens (*Rice, 2008*). With the general rise of antibiotic-resistant isolates, phage therapy has garnered some interest as an alternative to treat these infections (*Qin et al., 2022*; *Yin et al., 2024*). However, biofilm formation frequently interferes with phage infection (*Luthe et al., 2023*), and even though a few bacteriophage that can target *P. aeruginosa* in a biofilm have been described (*Manohar et al., 2024*; *Knezevic et al., 2021*), the mechanism by which they infect these biofilm bacteria is unknown.

The extracellular matrix of *P. aeruginosa* biofilms is comprised of exopolysaccharides, including Psl, Pel, and alginate, as well as proteins and DNA (*Ma et al., 2022*; *Ma et al., 2009*). Psl is of significant interest, since it is critical for biofilm formation, where it is needed for the initial surface attachment (*Byrd et al., 2010*), as well as structural stability of the mature biofilm (*Colvin et al., 2012*). Psl has been detected on the surface of individual *P. aeruginosa* bacteria in an apparent helical pattern (*Ma et al., 2009*). It is also deposited on surfaces by a subset of motile explorer bacteria during the early stages of aggregate formation (*Zhao et al., 2013*). Psl production interferes with complement deposition and neutrophil functions, such as phagocytosis and ROS production (*Mishra et al., 2012*). Moreover, Psl enhances the intracellular survival of phagocytosed *P. aeruginosa*, as well as survival in mouse models of lung and wound infection (*Mishra et al., 2012*).

Here, we describe the discovery of a bacteriophage that uses Psl, this crucial biofilm exopolysaccharide, as a receptor. Interestingly, this bacteriophage only infects a subpopulation of planktonically growing *P. aeruginosa*, but it can disrupt biofilms and replicates efficiently on biofilm-grown bacteria. Moreover, the phage can reduce the bacterial burden in a corneal infection model, which involves the formation of a biofilm.

## Results

### Phage Clew-1 can form plaques on a *ΔfliF* mutant, but not wild-type *P. aeruginosa*

We screened wastewater samples at the three Northeast Ohio Regional Sewer District water treatment plants in Cleveland for bacteriophage. The majority of phages in these samples used type IV pili as a receptor, and we wanted to exclude these from our screen. We had previously generated a *ΔfliF ΔpilA* double mutant strain in the lab and decided to use it to exclude both surface appendages as potential receptors. This turned out to be fortuitous, since, surprisingly, the screen identified four phages that could form plaques on the *ΔfliF ΔpilA* double mutant strain, but not the parental wild-type *P. aeruginosa* PAO1. We named these Cleveland wastewater-derived phages Clew-1,–3, –6, and –10. Subsequent tests determined that it was the *fliF* deletion that rendered *P. aeruginosa* permissive for infection by these phages. All four Clew phage can plaque on a *fliF* deletion mutant of *P. aeruginosa* PAO1, but not the corresponding wild-type strain or *ΔpilA* mutant strain (*Figure 1A*, *Figure 1— figure supplement 1A, B*, *Figure 1—figure supplement 2*). An unrelated Pbunavirus we isolated in the same screen, which uses O-antigen as receptor (*Figure 1—figure supplement 3*), was used as

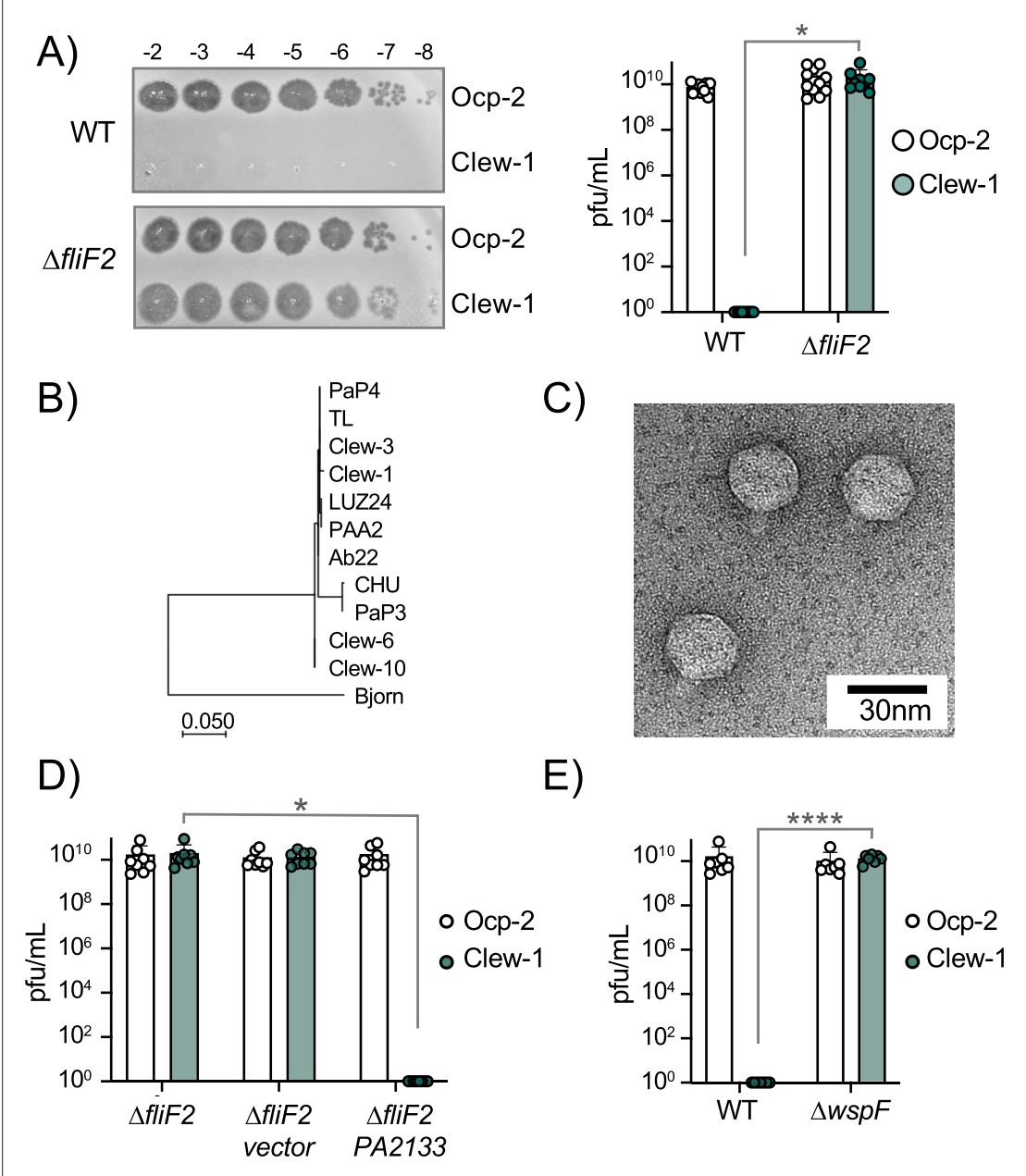

**Figure 1.** c-di-GMP levels control infection of *P. aeruginosa* by bacteriophage Clew-1. (**A**) Efficiency of plating experiment in which 3 µL of a 10x dilution series of bacteriophage Ocp-2 or Clew-1 were spotted on wild-type PAO1F, or PAO1F *ΔfliF2*. The adjacent graph shows the compiled results from 11 experiments. (**B**) Maximum likelihood phylogenetic tree of Clew-1 relative to other Bruynogheviruses (including the type phage, LUZ24) and phage Bjorn as an outgroup. Branch lengths are measured in number of substitutions per site in the terminase large subunit. (**C**) transmission electron micrograph of the Clew-1 phage. (**D**) Efficiency of plating experiment as in (**A**) assaying the effect of expressing the phosphodiesterase PA2133 from a plasmid. (**E**) Efficiency of plating experiment assaying the effect of deleting *wspF* on Clew-1 resistance. (\*p<0.05, \*\*\*\*p<0.0001 by Student's t-test (**A**, **E**) or one-way ANOVA with Šídák's multiple comparisons test (**D**)).

The online version of this article includes the following source data and figure supplement(s) for figure 1:

**Source data 1.** Plating efficiency data for all replicate experiments in panel A.

**Source data 2.** Plating efficiency data for all replicate experiments in panel D.

**Source data 3.** Plating efficiency data for all replicate experiments in panel E.

**Figure supplement 1.** Phage Clew-3, Clew-6, and Clew-10 plaque on a fliF mutant of *P. aeruginosa*.

**Figure supplement 2.** Complementation of the *ΔfliF2* mutant.

*Figure 1 continued on next page*

**Figure supplement 3.** Efficiency of plating analysis on LPS mutant strains.

**Figure supplement 4.** Impact of flagellar mutations on bacteriophage Clew-1 sensitivity.

a control in these experiments (<u>O</u>ur <u>c</u>ontrol <u>p</u>hage, Ocp-2). The Clew bacteriophages belong to the family of Bruynogheviruses (*Lefkowitz et al., 2018*) and are all highly related (*Figure 1B*, *Figure 1—figure supplement 1C*). Morphologically, like other members of the family, they are Podoviruses (*Figure 1C*).

## c-di-GMP levels control infection of *P. aeruginosa* by bacteriophage Clew-1

We next examined what part of the flagellum is involved in determining sensitivity to the Clew-1 phage. Mutations affecting the MS-ring (Δ*fliF*) and associated proteins FliE or FliG (*Minamino et al., 2020*) resulted in Clew-1 sensitivity. However, mutations in the ATPase complex only conferred partial sensitivity, and mutations affecting the hook or flagellar filament did not result in sensitivity, nor did a mutation that affects the type III secretion function of the flagellar basal body by impeding proton flux, *flhA(R147A)*(*Erhardt et al., 2017*; *Figure 1—figure supplement 4*). We, therefore, conclude that it is the presence of the MS-ring and not other aspects of the flagellum, such as assembly of the full flagellar structure or flagellar rotation, that controls phage sensitivity.

Interestingly, we found that deletion of *fleQ*, which is required for transcription of flagellar genes (*Arora et al., 1997*), had a very minor effect on Clew-1 phage susceptibility of the wild-type bacteria, and actually decreased Clew-1 susceptibility of the Δ*fliF* mutant bacteria (*Figure 1—figure supplement 1E*). FleQ is a c-di-GMP-responsive transcription factor that, among other things, reciprocally controls flagellar gene expression and production of biofilm-related characteristics, such as the production of the extracellular polysaccharides Psl and Pel, as well as the adhesin CdrA (*Arora et al., 1997*; *Oladosu et al., 2024*; *Dasgupta et al., 2003*). We, therefore, examined whether manipulating c-di-GMP levels controls phage susceptibility. To this end, we produced the c-di-GMP phosphodiesterase PA2133 from a plasmid (*Hickman et al., 2005*) to artificially lower c-di-GMP levels in the Δ*fliF2* deletion mutant strain. Conversely, we artificially elevated c-di-GMP levels in the wild-type by deleting the *wspF* gene (*Hickman et al., 2005*). Lowering c-di-GMP levels in the Δ*fliF2* mutant restored Clew-1 resistance (*Figure 1D*), whereas deleting *wspF* rendered the parental PAO1 strain phage sensitive (*Figure 1E*). Taken together, these data demonstrate that Clew-1 susceptibility is controlled by intracellular c-di-GMP levels and argue that absence of the MS-ring controls phage susceptibility through an increase in c-di-GMP.

## Phage Clew-1 requires Psl for infection

To better understand the host factors that control susceptibility and resistance to Clew-1 infection, we carried out a pair of TnSeq experiments. In the first of these, we mutagenized the wild-type strain PAO1F with the mini-mariner transposon TnFAC (*Wong and Mekalanos, 2000*), and the resultant mutant library was infected with phage Clew-1 at an MOI of 10 for 2 hr. The surviving bacteria were allowed to grow up after plating on an LB plate and the transposon insertion sites for the input and output pools were determined by Illumina sequencing. We identified insertion mutants that were depleted after infection (*Figure 2A*). Two of the genes with the most significant depletion were *fliF* and *fliG*, consistent with our previous analysis indicating that these mutations sensitize PAO1 to Clew-1 infection. Interestingly, we also noted depletion of *pch* and *bifA* insertions, both encoding phosphodiesterases that are involved in depleting c-di-GMP in the flagellated daughter cell after cell division (*Kulasekara et al., 2013*; *Laventie et al., 2019*; *Manner et al., 2023*; *Kuchma et al., 2007*). In fact, *pch* interacts with the chemotactic machinery (*Kulasekara et al., 2013*), highlighting, here too, the importance of c-di-GMP in controlling Clew-1 sensitivity.

In a reciprocal experiment, we carried out the TnSeq analysis in a *fliF* mutant strain. This analysis identified insertions in the *psl* operon as the most highly enriched group of mutants after Clew-1 selection, suggesting that Psl is required for phage infection (*Figure 2B*). We examined the requirement for Psl explicitly by generating *pslC* and *pslD* mutants in the PAO1F Δ*fliF2* strain background. PslC is a glycosyltransferase required for Psl biosynthesis, while PslD is required for Psl export from

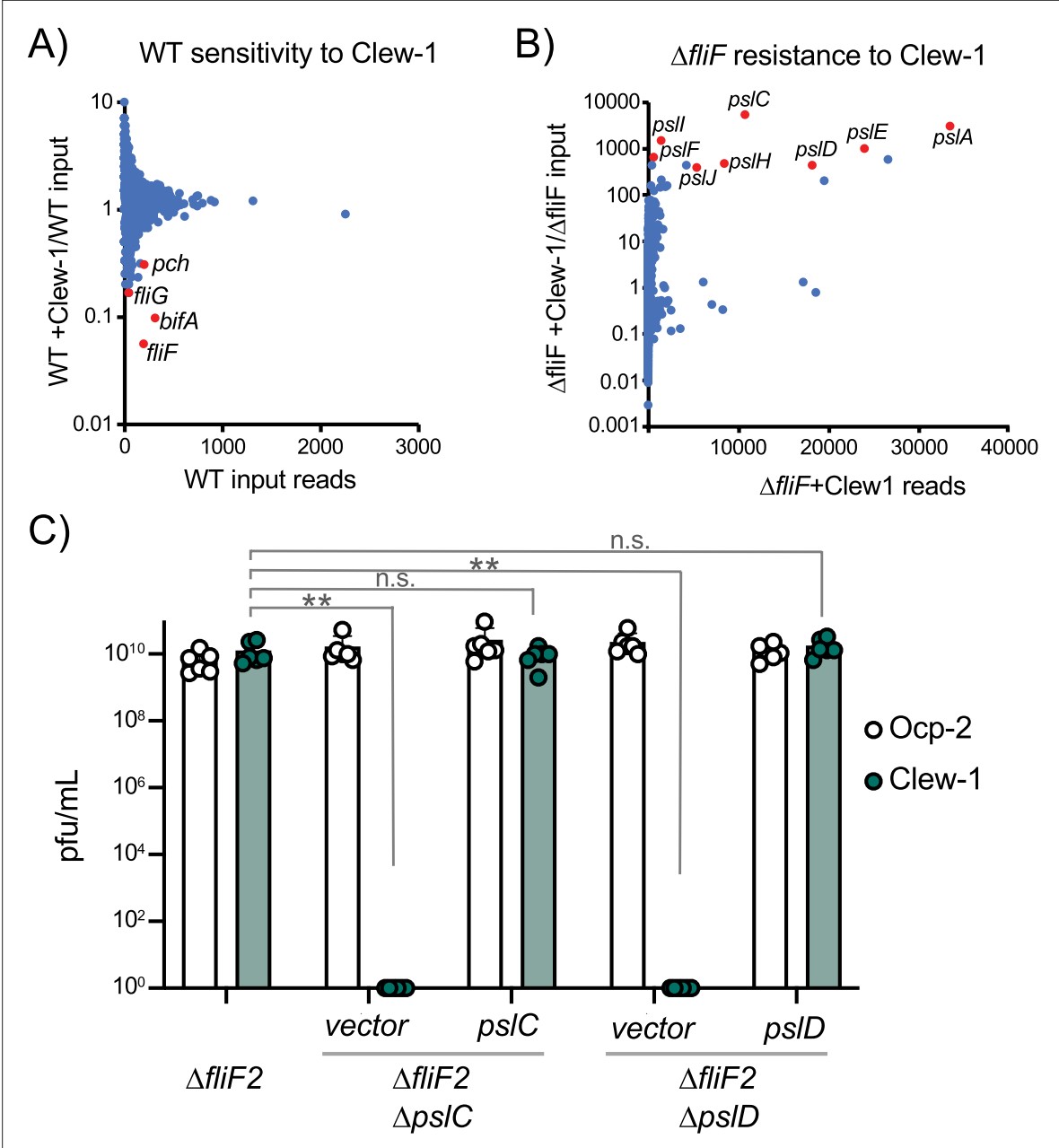

**Figure 2.** Bacteriophage Clew-1 uses Psl as a receptor to infect *P. aeruginosa*. (**A**) TnSeq experiment in which a pool of mariner transposon mutants of strain PAO1F was infected with phage Clew-1 for 2 hr. The number of insertions in the output pool was plotted against the ratio of the output and input pool. (**B**) Similar TnSeq analysis as in (**A**) but using PAO1F *ΔfliF2*. (**C**) Efficiency of plating analysis on *ΔfliF2 ΔpslC* and *ΔfliF2 ΔpslD*, Psl biosynthesis mutants, either harboring an empty vector or a complementing plasmid (n=6). Clew-1 values were compared by one-way ANOVA with Šídák's multiple comparisons test (**p<0.01, n.s. not significant).

The online version of this article includes the following source data for figure 2:

**Source data 1.** Plating efficiency data for all replicate experiments in panel C.

the cell (***Byrd et al., 2009***; ***Wu et al., 2019***). Deletion of either *pslC* or *pslD* rendered the *fliF* mutant bacteria Clew-1 resistant and sensitivity could be restored through complementation using a plasmid-borne copy of the deleted open reading frame (***Figure 2C***). These data demonstrate that Psl production is required for infection of *P. aeruginosa* by phage Clew-1.

## Phage Clew-1 attachment is Psl-dependent

We next examined whether attachment of Clew-1 to *P. aeruginosa* is Psl-dependent. We first used efficiency of the center of infection (ECOI) analysis to examine attachment. In this analysis, the phage is allowed to adhere to the bacteria for 5 min, before washing the bacteria to remove unattached phage. The bacteria are then diluted, mixed with top agar and a sensitive indicator bacterium (*ΔfliF2*), and then plated to allow for plaque formation as a biological readout of attached bacteriophage. Attachment of phage Clew-1 is Psl-dependent. Interestingly, we were able to detect Psl-dependent attachment both with wild-type and *ΔfliF2* mutant bacteria (*Figure 3A*), which contradicted our initial efficiency of plating experiments. We, therefore, reexamined phage susceptibility by monitoring phage infection in liquid media and generating lysis curves for wild-type and *ΔfliF2* mutant bacteria, as well as their *ΔpslC* mutant derivatives (*Figure 3—figure supplement 1*). The *ΔfliF2* mutant strain was lysed after ~40 min of infection. The wild-type bacteria displayed a significant slowing of growth upon Clew-1 infection when compared to the uninfected culture, but not clear lysis as was observed with the *ΔfliF2* mutant. In both instances, deleting *pslC* abolished any phage-dependent effect on growth.

We hypothesized that perhaps the difference between wild-type and *ΔfliF2* mutant bacteria is due to the fraction of cells that are producing Psl and therefore permissive for phage attachment. To test this hypothesis, we labeled phage Clew-1 with the DyLight594 fluorescent dye and examined attachment directly by microscopy (*Figure 3B*). We observed a statistically significant increase in the percentage of bacteria with attached bacteriophage in the *ΔfliF2* mutant bacteria compared to the wild-type, arguing that the increase in c-di-GMP in the *ΔfliF2* mutant increases the fraction of Clew-1 susceptible cells in the population. As anticipated, no phage was observed attached to the corresponding *ΔpslC* mutant (*Figure 3C*, *Figure 3—figure supplement 2*).

## Phage Clew-1 binds to Psl directly

We next examined whether phage Clew-1 can bind to Psl directly. We first determined whether we could precipitate phage Clew-1 from filter-sterilized culture supernatants of a *ΔfliF2* mutant using an antibody directed against Psl. The presence of the phage was determined by quantitative PCR. We were able to pull down phage Clew-1 in a Psl and antibody-dependent manner with *ΔfliF2 ΔpslC* culture supernatants serving as a control (*Figure 4A*). Notably, we observed some Psl-dependent attachment in the absence of antibody, arguing that Psl binds non-specifically to the magnetic beads we used in our experiments. Including the anti-Psl antibody resulted in a statistically significant increase compared to this background level of attachment (*Figure 4A*). We next repeated the pull-down using a partially purified fraction of cell-associated Psl to repeat the pulldown and again found Psl and antibody-dependent precipitation of phage Clew-1 (*Figure 4—figure supplement 1*). Finally, we examined phage binding using a biotinylated, affinity-purified preparation of Psl and found that we could pull down the phage using this Psl fraction as well, arguing that Clew-1 binds Psl directly (*Figure 4B*).

## Phage Clew-1 infects wild-type *P. aeruginosa* in biofilms

Since Clew-1 exploits Psl for infection and Psl is a key component of the biofilm matrix of most strains of *P. aeruginosa* (*Tabor et al., 2018*; *Zegans et al., 2016*), we hypothesized that, perhaps, Clew-1 can infect biofilm bacteria. We used a static biofilm model to test this hypothesis. Biofilms were established overnight in a 96-well plate. One plate was washed and fixed with ethanol to quantify the day one biofilm mass using crystal violet staining. In a second plate, established in parallel, the biofilms were washed with PBS and LB was added back, either without addition or with 10^9 pfu of phage Clew-1 or phage Ocp-2. The plates were incubated overnight and the next day, the day 2 biofilm mass was quantified using crystal violet. A similar experiment was carried out in 5 mL culture tubes to illustrate the result is shown in *Figure 5A*. The averages of five biological replicates in the 96-well experiment are shown in *Figure 5B*. Treatment of the day one biofilm with phage Clew-1 resulted in a statistically significant decrease in biofilm mass compared to the biomass present at day 1. Phage Ocp-2 infection, on the other hand, did not result in a reduction in biofilm (*Figure 5B*). Notably, phage Clew-1 was not able to reduce a biofilm formed by *P. aeruginosa* strain PA14, a natural *psl* mutant (*Figure 5—figure supplement 1*).

To corroborate this result, we also conducted a converse experiment where we monitored the ability of phage Clew-1 or Ocp-2 to replicate on biofilm bacteria over a 2 hr period. Here, biofilms

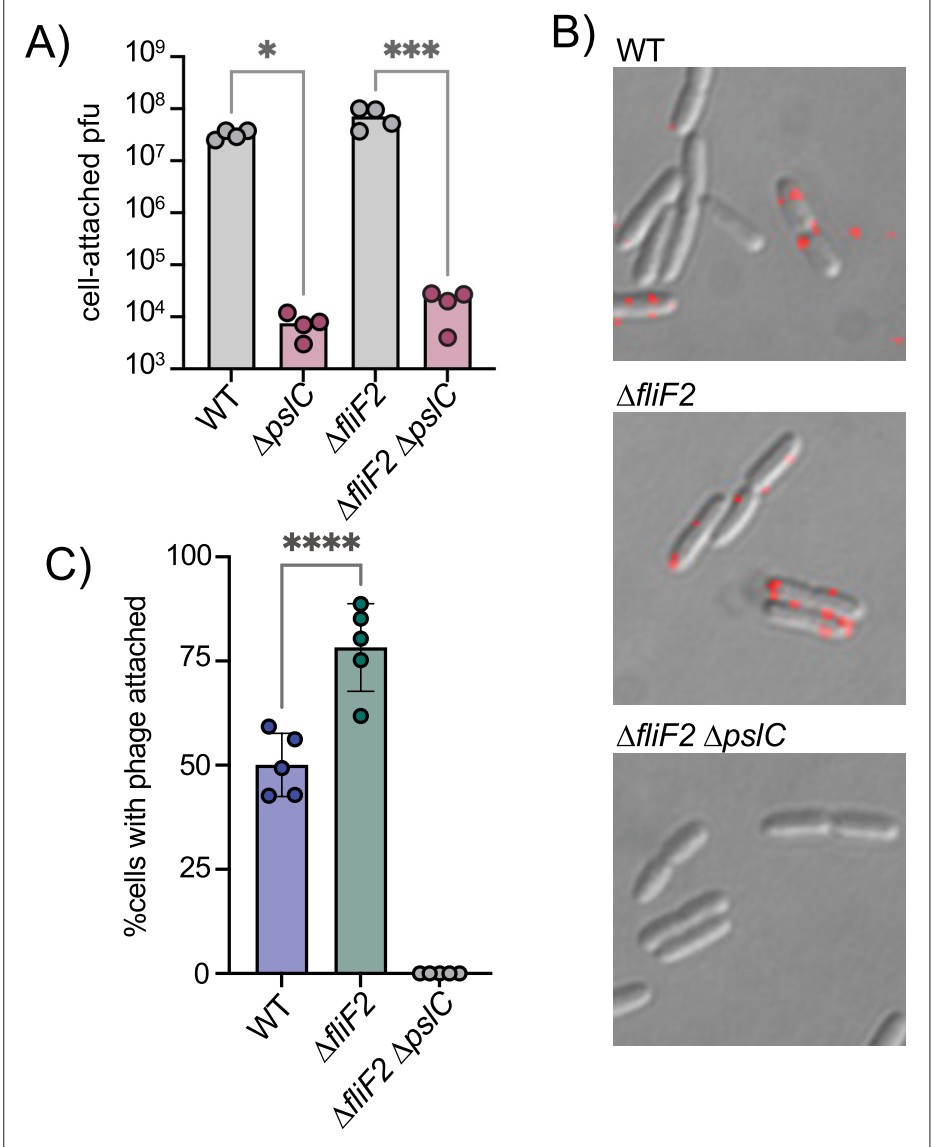

**Figure 3.** The *ΔfliF2* mutation changes the fraction of cells that phage Clew-1 binds to. (**A**) Efficiency of center of infection analysis. The indicated strain was infected for 5 min at an MOI of 0.01, the bacteria were pelleted, washed 3x with PBS, then diluted and mixed with an excess of the *ΔfliF2* mutant strain, top agar, and plated on an LB agar plate. The number of plaques was used to calculate the number of phages that attached and productively infected the initial strain. (**B**) Phage Clew-1 was labeled with DyLight594 fluorophores, bound to the indicated wild-type or mutant bacteria (15 min in LB), washed, and fixed with paraformaldehyde. Phages attached to bacteria was imaged by fluorescence microscopy and attachment was quantified over five biological replicates, shown in (**C**). Attachment was compared by one-way ANOVA with Šídák's multiple comparisons test. *p<0.05, ***p<0.001,****p<0.0001.

The online version of this article includes the following source data and figure supplement(s) for figure 3:

**Source data 1.** Efficiency of Center of Infection, viral titer data for all replicate experiments in panel A.

**Source data 2.** Phage attachment data for all replicate experiments shown in panel C.

**Figure supplement 1.** Lysis curves following bacteriophage Clew-1 infection.

**Figure supplement 2.** Distribution of phage attached to bacterial cells.

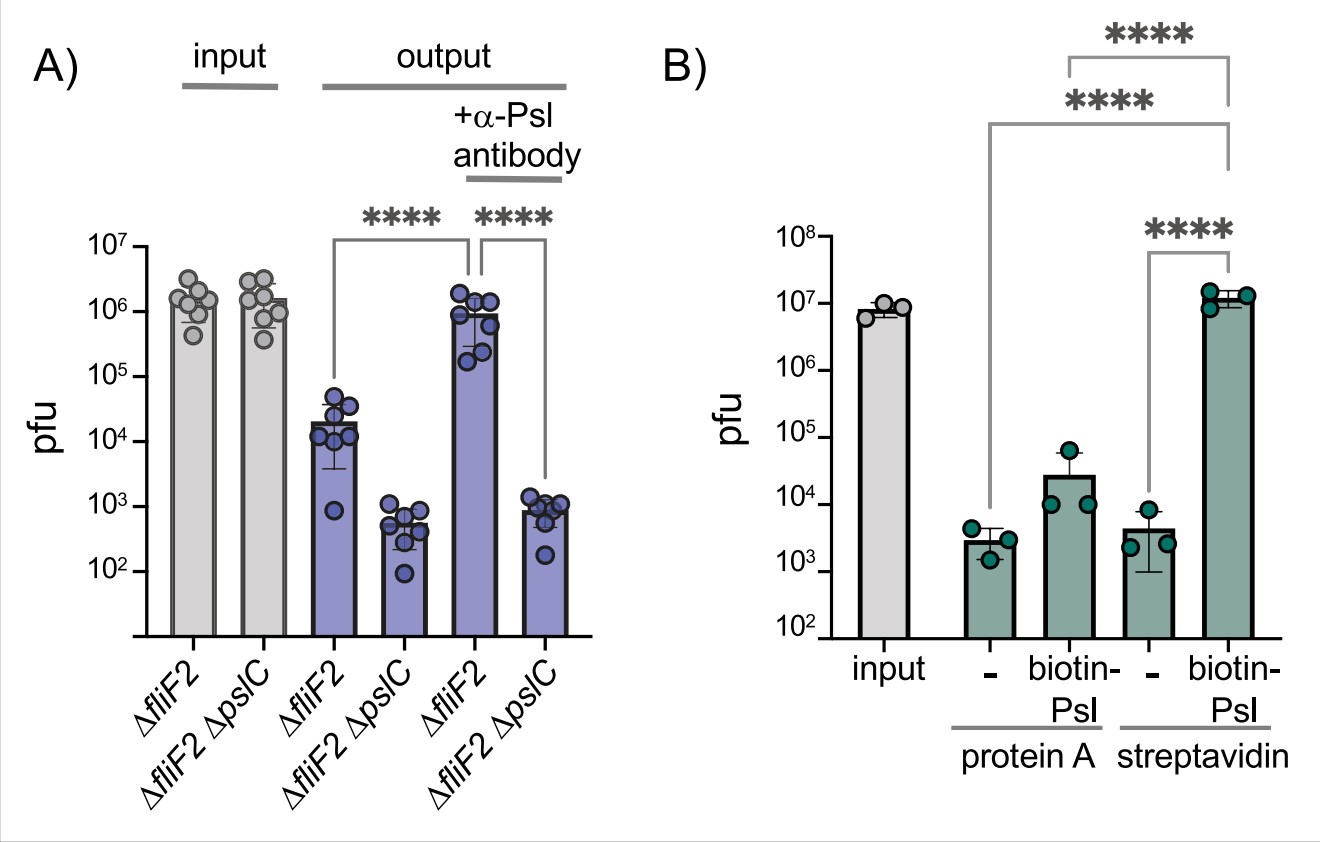

**Figure 4.** Phage Clew-1 binds to Psl. (**A**) Sterile filtered mid-log culture supernatants of PAO1F *ΔfliF2* or PAO1F *ΔfliF2 ΔpslC* were incubated with phage Clew-1, as well as magnetic protein-A beads and, where indicated, a rabbit, anti-Psl antiserum. Beads were collected, washed 3x, and phage in the input and output samples were quantified by qPCR (seven independent replicates). (**B**) Phage Clew-1 was incubated for 1 hr in SM buffer with affinity-purified, biotinylated Psl (biotin-Psl) and magnetic protein A beads, or magnetic streptavidin beads (SA), where indicated. Beads were collected and washed 3x, and phage in the input and output samples were quantified by qPCR (three independent replicates). Statistical significance was determined by ANOVA with Sidák post-hoc test (****p<0.0001).

The online version of this article includes the following source data and figure supplement(s) for figure 4:

**Source data 1.** All replicate data of viral titers (measured by qPCR) in the pull-down experiment in panel A.

**Source data 2.** All replicate data of viral titers (measured by qPCR) in the pull-down experiment in panel B.

**Figure supplement 1.** Phage Clew-1 binds to partially purified, cell-associated Psl.

**Figure supplement 2.** Phage Clew-1 does not degrade Psl.

were generated overnight in 5-mL culture tubes, the biofilms were washed with PBS and exposed to 10^5 pfu/mL phage Clew-1 or phage Ocp-2 for 2 hr. At the end of the experiment, the culture supernatants were filter-sterilized and the phages were titered. Consistent with the reduction in biofilm mass seen in *Figure 5B*, we found that phage Clew-1, but not Ocp-2, was able to replicate when grown on biofilm bacteria (*Figure 5C*).

In order to determine whether Clew-1 infection results in decolonization of bacteria from the biofilm or killing of biofilm bacteria, we carried out two complementary experiments. In the first experiment, we generated wild-type *P. aeruginosa* biofilms on polystyrene microbeads grown overnight with agitation. The beads were then washed with PBS and incubated for 2 hr either in LB or LB with Clew-1. Bacteria were released from the biofilm by sonication and the input CFU (total CFU attached to beads) were compared to the output after 2 hr of incubation (sum of supernatant and bead-associated viable bacteria). Clew-1 resulted in a significant reduction in viable CFU, arguing that phage infection kills biofilm bacteria (*Figure 5D*). In a related experiment, we used LIVE/DEAD staining of bacteria to assess bacterial viability by microscopy. We established biofilms in microwell slides overnight, then added Syto 9 and propidium iodide (PI) to each well along with phage Clew-1, or a buffer control,

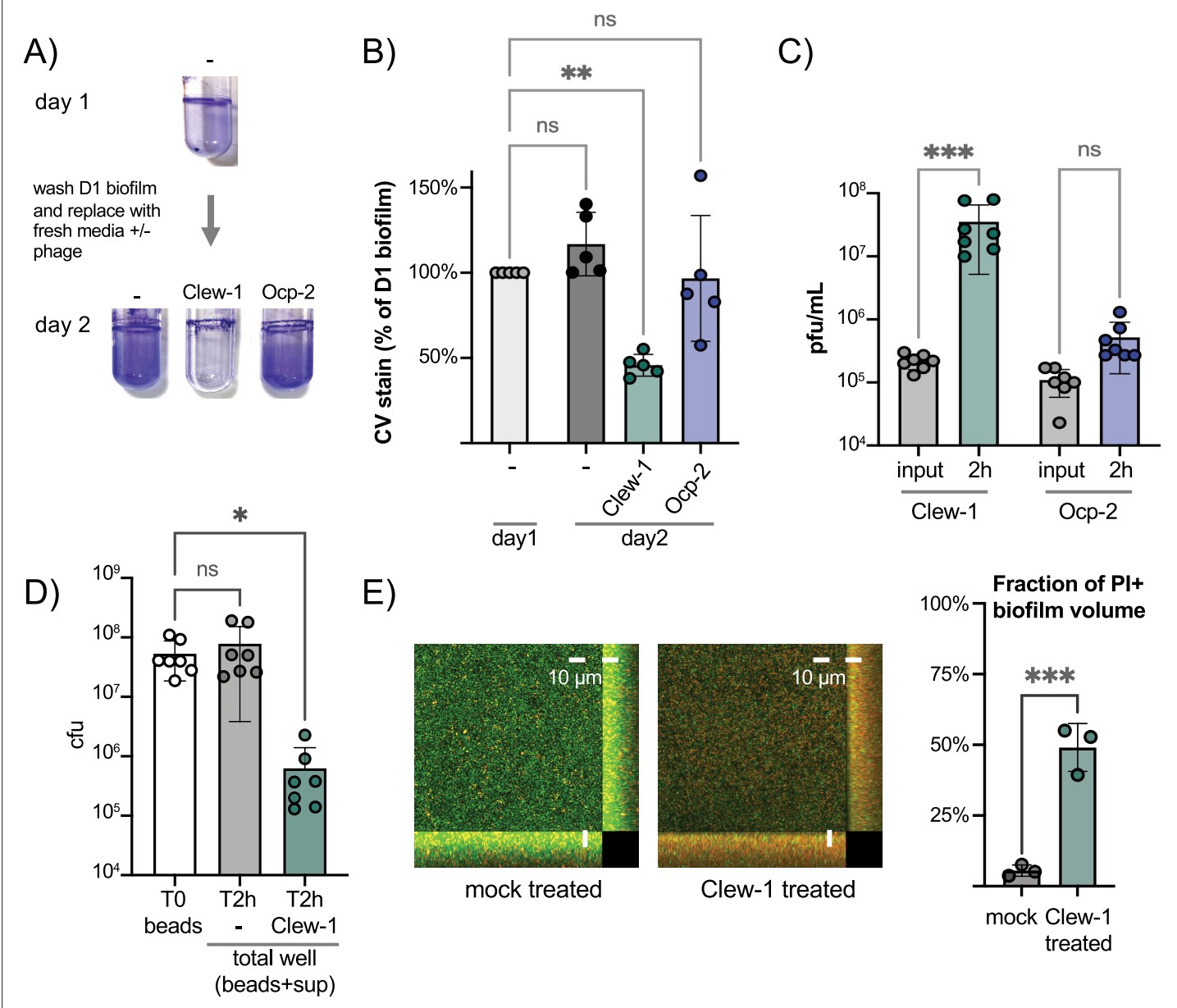

**Figure 5.** Phage Clew-1 can infect *P. aeruginosa* in biofilms. (**A**) Biofilms of wild-type *P. aeruginosa* PAO1F were established overnight in 5 mL culture tubes (1 mL culture), the tubes were washed with PBS and 1.2 mL LB, LB with 10^10 pfu phage Clew-1, or LB with 10^10 pfu phage Ocp-2 were added back to each tube (one was fixed with EtOH to represent the 1-day-old biofilm). The following day all biofilms were washed with PBS and stained with crystal violet. (**B**) PAO1F biofilms were established overnight in a 96-well plate (150 µL of culture, six technical replicates/condition), washed, and incubated overnight with 200 µL of LB or LB with 10^9 pfu bacteriophage Clew-1 or Ocp-2. The biofilms were then washed, fixed, and stained with crystal violet, which was then quantified spectrophotometrically. The day 1 controls were set to 100% (five biological replicates). (**C**) Growth of phage Clew-1 or Ocp-2 was assayed by establishing a static biofilm in 5 mL culture tubes overnight. The biofilms were washed with PBS, then LB with 10^5 pfu/mL of phage Clew-1 or Ocp-2 was added back. Biofilms were incubated at 37°C for 2 hr 15 min, the culture supernatants were filter sterilized, and input and output phage concentrations were titered (6 biological replicates). (**D**) Wild-type PAO1 biofilms were grown overnight, with agitation, on 6 mm polystyrene beads. Biofilms were rinsed with PBS and incubated with LB or LB with 10^9 pfu Clew-1 for 2 hr. The sum of bead-associated and supernatant CFU was titered at the 2 hr mark and compared to the bead-associated biofilm input (T0) to assess total viable CFU (seven biological replicates). (**E**) Biofilms were grown overnight in 8-well slides, at which point the LIVE/DEAD stain dyes, Syto 9 and propidium iodide were added, either on their own (mock) or in the presence of Clew-1. After another 24 hr of incubation, biofilms were imaged by confocal microscopy. Maximum intensity projections of the collected Z-stacks for one replicate are shown, along with YZ and XZ projections to the right and bottom of the image, respectively. The fraction of the total biofilm volume in the image stack that is propidium iodide positive was determined using BiofilmQ (three biological replicates).

*Figure 5 continued on next page*

*Figure 5 continued*

Statistical significance was determined by ANOVA with Šídák's multiple comparisons test, except for panel (**E**), where a two-tailed, unpaired t-test was applied (*p<0.05, **p<0.01, ***p<0.001).

The online version of this article includes the following source data and figure supplement(s) for figure 5:

**Source data 1.** Replicate data for biofilm biomass as measured by crystal violet retention, shown in panel B.

**Source data 2.** Replicate data for viral titers grown on biofilms, shown in panel C.

**Source data 3.** Replicate data for bacterial titers after phage treatment of biofilms, shown in panel D.

**Source data 4.** Replicate data of the fraction of the biofilm biovolume that stains positive for propidium iodide, shown in panel E.

**Figure supplement 1.** *P. aeruginosa* PA14 biofilms are insensitive to phage Clew-1.

and incubated the slide overnight again before imaging the biofilms by confocal microscopy. Infection with Clew-1 resulted in a significant increase in the fraction of the biofilm volume that is PI-positive, indicating that the biofilm bacteria are dying (*Figure 5E*).

## Phage Clew-1 can clear *P. aeruginosa* in a mouse keratitis model

Given the ability of phage Clew-1 to infect *P. aeruginosa* biofilms, we next examined whether Clew-1 could be used to treat a *P. aeruginosa* infection. Corneal infections by *P. aeruginosa* involve the formation of a biofilm (*Thanabalasuriar et al., 2019*; *Zegans et al., 2016*). In fact, a bivalent antibody directed against Psl and the type III secretion needle-tip protein, PcrV, was found to be effective in clearing such corneal infections (*Thanabalasuriar et al., 2019*). Moreover, multidrug-resistant *P. aeruginosa* was the cause of a recent outbreak of blinding corneal infections in the USA that was associated with contaminated eye drops (*Morelli et al., 2023a*; *Morelli et al., 2023b*). We, therefore, examined the ability of Clew-1 to reduce the *P. aeruginosa* bacterial burden in a clinically relevant murine model of blinding corneal disease. Mice were infected with 5*10^4 cfu of wild-type *P. aeruginosa* strain PAO1 and given a topical application of 5*10^9 pfu Clew-1 in 5 µL of PBS, or PBS alone, at 24 hr and 48 hr post-infection (*Figure 6A*). After 48 hr of infection, we quantified corneal opacity, a measure that correlates with the infiltration of immune cells (*Sun et al., 2010*; *Hazlett et al., 1992*; *Lee et al., 2003*), and GFP fluorescence (produced by the *P. aeruginosa* strain used in the infection) by image analysis. We also assessed the bacterial burden (colony- forming units). Mice infected with PAO1 developed severe corneal disease manifest as corneal opacification in the region of bacterial growth indicated by GFP fluorescence (*Figure 6B–D*). In contrast, phage Clew-1 treatment significantly reduced the bacterial burden, and in some instances, completely eradicated the infection, as measured by GFP fluorescence and CFU (*Figure 6D and E*). Corneal opacity was similarly reduced by the phage treatment, indicating that inflammation was also starting to resolve (*Figure 6B and C*).

## Discussion

We describe the isolation of four phages belonging to the family of Bruynogheviruses that use the *P. aeruginosa* exopolysaccharide Psl as a receptor. Psl is not a capsular polysaccharide, so this distinguishes the Clew phages from phages such as KP32 that infect *Klebsiella pneumoniae*. Moreover, these *Klebsiella* phages use a capsular depolymerase to break down the capsular polysaccharide (*Squeglia et al., 2020*; *Dunstan et al., 2021*). Clew-1, on the other hand, has no such activity (*Figure 4—figure supplement 2*) arguing that the role of Psl in infection is distinct from that seen in capsule-targeting bacteriophage.

Phage Clew-1 has the surprising quality that it fails to plaque on wild-type *P. aeruginosa* PAO1, but forms plaques on a *fliF* mutant. We determined that the *fliF* mutation generates a c-di-GMP-dependent signal that up-regulates Psl production. Importantly, it increases the fraction of bacteria to which the phage can bind, resulting in efficient lysis in liquid cultures, and plaque formation in top agar. Plaque formation is likely masked in the wild-type bacteria by the fraction of cells that are not phage-susceptible. Notably, certain Bruynogheviruses are able to bind to *P. aeruginosa* PAO1, but not plaque (*Costa et al., 2024*). We now have an explanation for this observation.

The identification of Psl as phage receptor prompted us to examine the ability of phage Clew-1 to infect wild-type *P. aeruginosa* in a biofilm. We found that phage Clew-1, unlike the unrelated Ocp-2 phage, was able to disrupt biofilms formed by wild-type bacteria. Moreover, Clew-1 was able to

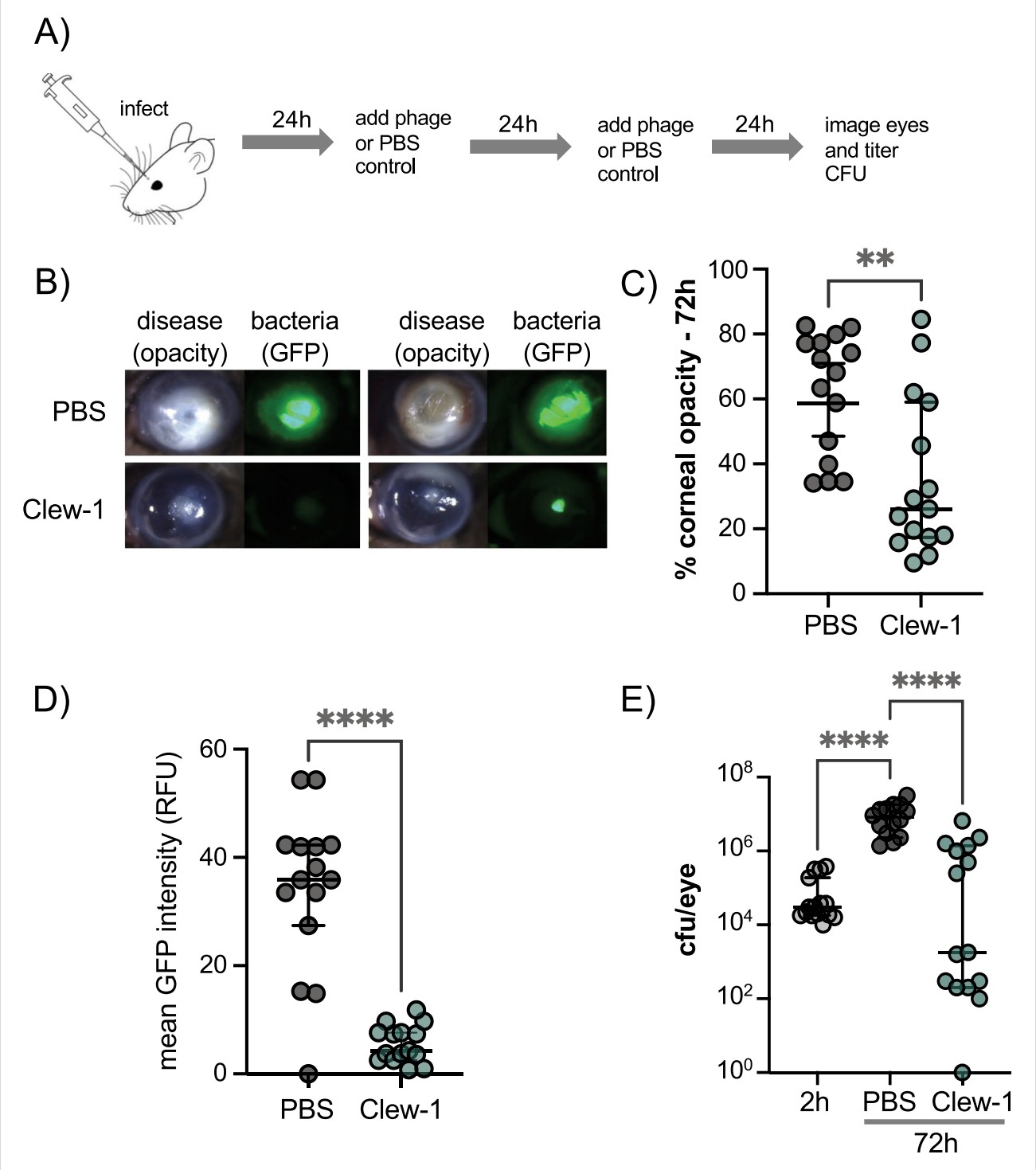

**Figure 6.** Phage Clew-1 reduces the bacterial burden in a mouse cornea model of infection. (**A**) Mouse corneas were abraded and infected with 5*10^4 cfu strain PAO1F/pP25-GFPo, which produces GFP constitutively. Infected corneas were treated with 2*10^9 pfu phage Clew-1 or a PBS control at 24 hr and 48 hr post-infection. (**B**) At 72 hr post-infection, the corneas were imaged by confocal microscopy to estimate the opacity (driven largely by the infiltration of neutrophils) and GFP fluorescence (produced by infecting *P. aeruginosa*). Representative images are shown in (**B**). Opacity and GFP fluorescence were quantified by image analysis and are graphed in (**C**) and (**D**), respectively. (**E**) Eyes were also homogenized and plated for CFU to determine the total bacterial burden at the end of the experiment. Significance was determined by Mann-Whitney test (**C**, **D**) or ANOVA with Kruskal-Wallis post-hoc test (**E**) (**p<0.01, ****p<0.0001).

The online version of this article includes the following source data for figure 6:

*Figure 6 continued on next page*

*Figure 6 continued*

**Source data 1.** Replicate data for corneal opacity, shown in panel C.

**Source data 2.** Replicate data for GFP fluorescence in infected corneas shown in panel D.

**Source data 3.** Replicate data for colony forming units recovered from infected corneas shown in panel E.

actively replicate on biofilm bacteria, while phage Ocp-2 could not. Taken together, our data suggest that phage Clew-1 has evolved to replicate in *P. aeruginosa* growing in a biofilm. Given the prevalence of bacterial biofilms in nature, this specialization makes sense. Moreover, our observation suggests that we may have underestimated the prevalence of biofilm-tropic bacteriophage since standard isolation techniques using plaque formation of wild-type bacteria would miss phage akin to Clew-1. In fact, another bacteriophage was recently described that requires an intact *psl* operon for replication and can only plaque on PAO1 with elevated c-di-GMP levels (*Manner et al., 2023*). This bacteriophage, Knedl, belongs to the family of Iggyviruses (*Maffei et al., 2024*), highlighting that more biofilm-tropic bacteriophage wait to be discovered. Our data also suggest that, for *P. aeruginosa*, using a Δ*fliF* Δ*pilA* double mutant will allow us to enrich for biofilm-specific bacteriophage, by excluding dominant type IV pilus-dependent phage and up-regulating biofilm-specific surface structures such as Psl, Pel, and CdrA. Given the importance of biofilms in contributing to the antibiotic resistance of *P. aeruginosa* in infections such as the CF lung, catheters, or wound infections, treatment modalities that are targeted towards biofilm bacteria are sorely needed. Indeed, phage Clew-1 shows some promise in this regard, since it was able to kill biofilm bacteria in vitro, and control *P. aeruginosa* infection in a mouse model of keratitis, which involves biofilm formation at the site of infection. While Clew-1 plaques can sometimes look turbid, this is likely a function of the fact that Clew-1 can only bind to a fraction of the bacterial population (80–90% in the case of PAO1 Δ*fliF2*, *Figure 3*). We have not been able to isolate Clew-1 lysogens, which is consistent with observations using other *Bruynogheviruses* (*Knezevic et al., 2021*; *Ceyssens et al., 2008*; *Alvi et al., 2021*) and argues that Clew-1 is lytic and not temperate. While many bacteriophages are not able to infect *P. aeruginosa* biofilms, some phages with the ability to target biofilms have been described, including the *Bruynoghevirus* Delta (*Knezevic et al., 2021*). We present here a way by which phages that target *P. aeruginosa* biofilms can be enriched during isolation.

Capsule-targeting bacteriophages tend to use LPS as a secondary receptor for tight binding to the target cell. While Clew-1 does not seem to require either A- or B-band LPS for infection (*Figure 1—figure supplement 3*), we did hit several genes involved in the biosynthesis of outer core LPS in our TnSeq analysis (*Supplementary file 3*). We tested whether one of these genes, *ssg* (PA5001), is required for Clew-1 infection and found that this is the case (*Figure 1—figure supplement 3*). How binding to Psl and LPS relates to the infection of *P. aeruginosa* by Clew-1 will be examined in a future study.

Another interesting aspect of the work described herein is the relationship between the presence of the MS-ring (FliF) and its associated proteins FliE and FliG with c-di-GMP levels. While it has been noted previously that flagellar mutations lead to increases in c-di-GMP levels and increased production of Psl upon surface contact (*Harrison et al., 2020*), our results differ somewhat in that phage susceptibility was primarily the result of loss of the MS-ring and associated proteins (FliEFG), not, for example, the flagellar filament (FliC). This may be due to differences between planktonically grown (as in our study) and surface-attached bacteria. Surface contact leads to up-regulation of c-di-GMP through surface sensing by the Wsp chemosensory system (*Hickman et al., 2005*; *Armbruster et al., 2019*). Attached bacteria divide asymmetrically, with c-di-GMP levels decreasing in the flagellated daughter cell (*Kulasekara et al., 2013*; *Laventie et al., 2019*; *Manner et al., 2023*; *Christen et al., 2010*). This asymmetry requires the phosphodiesterase Pch, which has been reported to bind to the chemosensory protein CheA (*Kulasekara et al., 2013*; *Laventie et al., 2019*). A second phosphodiesterase, BifA, is also required for maintaining c-di-GMP homeostasis and developing an asymmetric program of cell division upon attachment to surfaces (*Manner et al., 2023*; *Kuchma et al., 2007*). In our TnSeq experiment, we found that insertions in flagellar genes, such as *fliF* and *fliG*, but also insertions in *pch* and *bifA* resulted in Clew-1 sensitivity. Whether the strong Clew-1 sensitivity associated with deletion of *fliF*, *fliE*, or *fliG* in our data, relative to deletions in other flagellar components, relates to a pivotal role of the MS-ring in controlling the activity of Pch and/or BifA is unclear, but worth

further investigation. However, our work, along with the work of the Jenal group (*Manner et al., 2023*), suggests that phages such as Clew-1 or Knedl could be a useful tool for interrogating c-di-GMP signaling pathways in *P. aeruginosa*.

In summary, we have described here the isolation of a group of bacteriophages that target *P. aeruginosa* biofilms by using the exopolysaccharide Psl as a receptor. Consistent with the critical role of Psl as part of the *P. aeruginosa* biofilm matrix, we demonstrate that phage Clew-1 can replicate on biofilm bacteria and control *P. aeruginosa* in a mouse model of keratitis. Moreover, we have described a generalizable method that allows for the enrichment of biofilm-tropic bacteriophage, which is important due to their potential utility in combating biofilm infections that are notoriously recalcitrant to antibiotic therapy.

## Methods

### Strain construction and culture conditions

Bacterial strains were grown in LB (10 g/L tryptone, 5 g/L yeast extract, 5 g/L NaCl) at 37°C unless indicated otherwise. Bacterial strains and plasmids used in this study are listed in *Supplementary file 1*. Mutations were introduced into the genome of *P. aeruginosa* by allelic exchange. Briefly, flanks defining the mutation were amplified from the *P. aeruginosa* genome and cloned into plasmid pEXG2 by Gibson cloning. The oligonucleotides used for the amplifications were designed using AmplifX 2.1.1 by Nicolas Jullien (Aix-Marseille Univ, CNRS, INP, Inst Neurophysiopathol, Marseille, France - https://inp.univ-amu.fr/en/amplifx-manage-test-and-design-your-primers-for-pcr) and are noted in *Supplementary file 2*. Plasmid pEXG2 harboring the mutation construct was transformed into *E. coli* strain SM10 and mated at 37°C into *P. aeruginosa* by mixing the donor and recipient strains on an LB plate. The mating mixture was then plated on an LB plate with 30 μg/mL gentamicin and 5 μg/mL triclosan and grown overnight at 37°C (selecting against the *E. coli* donor strain). Cointegrates were restruck and subsequently grown in LB lacking salt until the culture was barely turbid. The bacteria were then plated on a sucrose plate (5% sucrose, 10 g/L tryptone, 5 g/L yeast extract) and incubated overnight at 30°C. Sucrose-resistant colonies were tested for gentamicin sensitivity and the presence of the mutation was tested by PCR.

Complementing plasmids were generated by amplifying the open reading frame and using Gibson assembly (*Gibson et al., 2009*) to clone it into pPSV37. The plasmids were then transformed into *P. aeruginosa* by electroporation.

All newly generated strains and plasmids are available from the corresponding author upon request.

For motility assays, individual bacterial colonies were used to inoculate motility agar plates (0.3% agar, LB plates) and incubated at 37°C for ~8 hr before imaging the plate.

### Bacteriophage isolation and sequencing

Bacteriophages were isolated from filter-sterilized wastewater samples from one of three Northeast Ohio Regional Sewer District wastewater treatment plants (Westerly, Southerly, or Easterly Sewage Treatment plant). Phages were isolated by picking individual plaques isolated on *P. aeruginosa* PAO1 Δ*fliF* Δ*pilA* and testing them for their ability to form plaques on the isolation strain or wild-type PAO1. Phage genome DNA was isolated as previously described (*Jakočiūnė and Moodley, 2018*). Genomic DNA was sequenced by Illumina sequencing and assembled using Spades version 3.15.3 (*Prjibelski et al., 2020*). Genomes were annotated using PhageScope (*Wang et al., 2024*). Genomes for Clew-1, Clew-3, Clew-6, and Clew-10, as well as Ocp-2, have been deposited in GenBank [accession# PQ790658.1, PQ790659.1, PQ790660.1, PQ790661.1, and PQ790662.1].

### CsCl purification of bacteriophage

Bacteriophages were purified by CsCl gradient based on a published protocol (*Piya et al., 2017*). Briefly, a 500 mL culture of PAO1F Δ*fliF2* was grown in LB to an $OD_{600}$ of ~0.2 and inoculated with phage Clew-1 or Ocp-2 at an MOI of 0.025. After about 4 hr of culture, the bacteria were pelleted (12,000 × g, 15 min, 4°C) and the supernatant was filtered through a 0.2μM filter. The supernatant was treated with DNAse and RNAse (1 μg/mL each) overnight at 4°C. The following day, the phages were pelleted by centrifugation (overnight, 7000 × g, 4°C), the supernatant discarded, and the pellets were resuspended in 1 mL of SM buffer (50 mM Tris Cl (pH 7.5), 100 mM NaCl, 8 mM $MgSO_4$) without BSA

each (~2 hr at 4°C). The concentrated phage prep was then spun at 12,000 × g for 10 min to pellet remaining cell debris. At this point, 0.75g of CsCl/mL was added to the cleared supernatant and the mixture was spun for 20 hr at 4°C at 32,000 rpm in Beckman Optima MAX-TL ultracentrifuge using an MLS-50 rotor to establish the gradient. The band with the phage was removed with a syringe and 20-gauge needle and transferred to a 3.5kDa cut-off dialysis cassette (Slide-A-Lyzer, Thermo). The phage prep was dialyzed overnight against SM, then 2x for 3 hr against SM, then overnight against PBS and 1x for 4 hr against PBS. The phage prep was tested for titer and, in the case of Clew-1, the ability to plaque on a ΔfliF2 mutant, but not wild-type PAO1F.

## Negative stain electron microscopy

The negative stain experiment was done as described previously (*Booth et al., 2011*). Briefly, a 3 μl phage Clew-1 sample (0.1–0.5 mg/ml) was loaded onto a glow-discharge carbon-coated grid for 60 s at room temperature and blotted with filter paper. The grid was touched with a water droplet and then blotted with filter paper. This process was repeated twice. The grid was then touched with a drop of 0.75% uranyl formate and blotted with filter paper. A second drop of 0.75% uranyl formate was applied to touch the grid for 30 s, blotted with filter paper, and then air dried before data collection. The images were taken by Tecnai T20 (FEI Company) equipped with a Gatan 4K × 4K CCD camera at 80,000x magnification.

## Efficiency of plating experiments

To test phage plating efficiency, bacterial strains were back-diluted 1:200 from overnight cultures and grown to early log phase (OD600 ~0.3). At this point, 50 μL of culture was mixed with 3 mL top agar (10 g/L tryptone, 5 g/L yeast extract, 5 g/L NaCl, 0.6% agar) and plated on an LB agar plate. Once solidified, 10-fold serial dilutions of the phage in SMB buffer (50 mM Tris Cl (pH 7.5), 100 mM NaCl, 8 mM MgSO$_4$, 0.1% bovine serum albumin) were spotted onto the agar using a multichannel pipette (3 μL spots). The spots were allowed to dry and the plates incubated overnight at 37°C.

## Efficiency of Center of Infection (ECOI) experiments

*P. aeruginosa* strains were grown to mid-logarithmic phase in LB supplemented with 5 mM MgCl2 and 0.1 mM MnCl2 (LBMM) concentrated and resuspended at a concentration of 10^9 cfu/mL in LBMM.

100 μL bacterial suspensions were infected at an MOI of 1 with phage Clew-1 (2 μL, 5*10^10 pfu/mL) for 5 min at 37°C, then pelleted (3' 10k RPM), washed 2x with 1 mL LB, and resuspended in 100 μL LB. The infected cells were serially diluted 10x, then 10 μL of diluted, infected bacteria (10^−4 for WT and ΔfliF2; 10^−1 for ΔpslC and ΔfliF2 ΔpslC) were mixed with 50 μL of the mid-log PAO1F ΔfliF2 culture and mixed with 2.5 mL top agar, plated on an LB plate, and incubated overnight at 37°C. The following day, plaques were counted to enumerate the cell-associated bacteria (*Hyman and Abedon, 2010*).

## TnSeq analysis

Strain PAO1F or PAO1F ΔfliF were mutagenized with transposon TnFac (*Wong and Mekalanos, 2000*), a mini-mariner transposon conferring gentamicin resistance. A pool of 3*10^6 (PAO1F) or 6*10^6 (PAO1F ΔfliF) insertion mutants was grown overnight, then diluted 1:200 in fresh LB and grown to an OD600 of 0.2. At this point, the bacteria were infected at an MOI of 10 with phage Clew-1 and incubated for 2 hr to allow infection and killing of susceptible bacteria. Bacteria from 1 mL culture were then pelleted, resuspended in 100 μL LB with 5 mM EGTA, and plated on 3 LB plates with 30 μg/mL gentamicin. The next day, surviving bacteria that had grown up were pooled and chromosomal DNA from the input and output pools was isolated using the GenElute Bacterial Genomic DNA Kit (Millipore-Sigma). Library preparation followed a published protocol (*Gallagher, 2019*). Genomic DNA was sheared to ~300bp using a Covaris focused ultrasonicator. The sheared DNA was repaired using the NEBNext End Repair Module (New England Biolabs) and subsequently tailed with a poly-dC tail using Terminal Transferase (New England Biolabs). Tailed chromosomal DNA fragments were amplified in two consecutive steps, using primers Mar1x and olj376 for the first round and Mar2-InSeq paired with a TdT_Index primer for the second round, based on the published protocol (*Gallagher, 2019*). The libraries were sequenced using an Illumina MiniSeq system using the transposon-specific primer MarSeq2. Reads with the correct Tn end sequence were mapped and tallied per site and per

gene using previously described scripts (*Gallagher, 2019*, and https://github.com/lg9/Tn-seq). The data (hits and # of reads for each gene) are listed for each strain and condition in *Supplementary file 3*.

## Clew-1 attachment by fluorescence microscopy

Bacteriophage Clew-1 was isolated from 500 mL of culture and purified using a CsCl gradient, following a protocol published by the Center for Phage Technology at Texas A&M University. After overnight dialysis of the phage into SM buffer, the phage was dialyzed three more times against PBS (2x for 3 hr and once overnight). The purified phage was titered by efficiency of plating analysis and labeled with a Dylight594 NHS-ester (Invitrogen) at a concentration of 0.2 mM, overnight in the dark. After labeling, the residual dye was removed by gel filtration using a Performa DTR gel filtration cartridge (EdgeBio) that had been equilibrated with PBS. The labeled phage preparation was titered to ensure that the phage concentration was unchanged and that the phage had not lost infectivity.

To assess phage attachment, wild-type PAO1F, PAO1F $\Delta fliF2$, or PAO1F $\Delta fliF2$ $\Delta pslC$ harboring plasmid pP25-GFPo, which directs the constitutive production of GFP, were grown in LB to an $OD_{600}$ of ~0.3–0.4, normalized to an $OD_{600}$ of 0.3, and 0.5 mL of the culture were infected for 10 min at 37°C with DyLight594-labeled Clew-1 phage at an MOI of 5. At this point, the infected bacteria were fixed with 1.6% paraformaldehyde [final concentration], incubated in the dark for 10 min, then the remaining paraformaldehyde was quenched through the addition of 200 µL of 1M glycine (10 min at RT). The bacteria were washed 3x with 500 µL of SM buffer and resuspended in 30 µL of SM buffer. 4 µL were spotted onto an agarose pad, covered with a coverslip, and imaged using a Nikon Eclipse 90i fluorescence microscope. Images were adjusted for contrast and false-colored using the Acorn software package (Flying Meat Software), and cell-associated bacteriophage were counted in ImageJ.

## Isolation and purification of Psl polysaccharide

Wild-type *P. aeruginosa* was grown for 18 hr in M63 minimal medium ($[NH_4]_2SO_4$, 2 g/l; $KH_2PO_4$, 13.6 g/l; $FeCl_3$, 0.5 mg/l, pH 7) supplemented with 0.5% Casamino acids (BD), 1 mM $MgCl_2$, and 0.2% glucose. Bacterial cells were removed by centrifugation, the supernatant lyophilized, and Psl isolated by affinity chromatography.

The affinity column was prepared by resuspending 0.286 g of CNBr-activated Sepharose (purchased from GE Healthcare; cat#17-0430-01) in 1 M HCl (1 mL). It was subsequently filtered and washed with 1 M HCl (60 mL) and coupling buffer (1.5 mL; 0.1 M NaHCO3, 0.5 M NaCl, pH = 9). The activated Sepharose was added to a solution of Cam-003 (*DiGiandomenico et al., 2012*) in coupling buffer (0.5 mL; 10 mg/mL) and was incubated for 2 hr at room temperature. The solvent was then removed by filtration, and the beads were washed with coupling buffer (3 × 1 mL). After removal of the solvent, the sepharose was incubated with blocking buffer (2 mL; 0.1 M Tris, 0.5 M NaCl, pH = 8.5) for 2 hr at ambient temperature. The beads were washed with wash buffer (4 mL) and coupling buffer (4 mL) for four cycles until the OD280 of the wash was <0.01. The derivatized beads were loaded onto a column and after washing with 5 column volumes of PBS buffer (pH = 7.4), the affinity column was ready to use.

Crude Psl (100 mg) was dialyzed (Thermo Scientific SnakeSkinTM Dialysis Tubing 3K MWCO) for three days and six exchanges of water and then concentrated to a final volume of 1 mL (10 mg/mL). It was loaded onto the affinity column and washed with PBS buffer (4 mL) in order to remove all not-retained material. Next, the captured Psl was eluted with glycine buffer (4 mL; 100 mM glycine × HCl, pH = 2.7). The glycine fraction was dialyzed (3 K MWCO) for three days and six exchanges of water and after lyophilization, pure Psl (80 µg) was obtained.

The solution was lyophilized, and the residue was fractionated by gel permeation chromatography on a Bio-Gel P-2 column (90 × 1.5 cm), eluted with 10 mM $NH_4HCO_3$. The collected fractions contained different sizes of Psl material: dimer (two repeating units), trimer (three repeating units), and high molecular weight polysaccharide. The high molecular weight polysaccharide fraction was used in our experiments.

Matrix-Assisted Laser Desorption/Ionization Time-of-Flight Mass Spectrometry experiments were performed using a Bruker ultrafleXtreme (Bruker Daltonics) mass spectrometer. All spectra were recorded in reflector positive-ion mode and the acquisition mass range was 200–6000 Da. Samples were

prepared by mixing the target 0.5 µL sample solutions with 0.5 µL aqueous 10% 2,5-dihydroxybenzoic acid as matrix solution.

## Precipitation of Clew-1 from culture supernatants and using purified Psl

For experiments in which binding of Clew-1 to Psl in culture supernatants was tested, PAO1F *ΔfliF2* or PAO1F *ΔfliF2 ΔpslC* were grown to mid-logarithmic phase, then the bacteria were pelleted and the supernatants filter sterilized using a 0.2µM filter. Culture supernatants were mixed with 1 µL of a rabbit, anti-Psl antibody (*Byrd et al., 2009*) as well as 10^7 pfu of phage Clew-1. The mixture was incubated on ice for 1 hr, then 10 µL of magnetic protein A beads (BioRad), washed 2x with SMB + 0.05% Triton X-100 (SMBT) were added to the mixture and incubated for an additional 30 min on ice. The magnetic beads were collected, washed 3x with SMBT and resuspended in 100 µl of SMBT. Presence of Clew-1 in input and output samples was determined by quantitative PCR using primers designed to amplify the tail fiber gene, gp12.

Experiments using partially purified, cell-associated Psl were carried out in SM buffer. 100 µL of SM buffer were mixed with 10^7 pfu of phage Clew-1, as well as 1µg of a partially purified, deproteinated fraction of cell-associated Psl (*Chiba et al., 2015*) and incubated for 1 hr on ice. All subsequent steps were the same as for the culture supernatants, above. Samples were resuspended in 100 µL SMBT before quantifying Clew-1 levels.

Experiments using affinity-purified, biotinylated Psl were carried out in SM. Here too, 10^7 pfu Clew-1 were incubated with 1µg of affinity-purified, biotinylated Psl. The samples were either incubated with streptavidin-coated Dynabeads (M280, Invitrogen) to precipitate the biotinylated Psl (or with magnetic protein A beads as a specificity control). Otherwise, the experiments were carried out as for the partially purified, cell-associated Psl fraction, above.

## Static biofilm experiments

Static biofilm experiments were carried out based on a published protocol (*O'Toole, 2011*). *P. aeruginosa* PAO1F was grown to mid-logarithmic phase in LB and then diluted to an OD600 of 0.05 and used to inoculate either 5 mL polystyrene tubes (1 mL) or six wells in a polystyrene 96-well plate (150 µL). The cultures were incubated overnight at 37°C in a humidified incubator with a 5% CO2 atmosphere. The following day, one set of biofilm samples was washed three times with PBS, fixed with 95% ethanol for 20 min, and subsequently air-dried after removing the ethanol. The remaining biofilm samples were washed 2x with PBS and reconstituted with pre-warmed LB (1.2 mL in 5 mL tube biofilms, 200 µL in 96-well plates), or LB harboring either 10^9 pfu of phage Clew-1 or phage Ocp-2. The biofilm samples were again incubated overnight at 37°C in a humidified incubator with a 5% CO2 atmosphere, and subsequently washed and fixed as the control samples, above. The fixed and dried biofilms were stained with a 0.1% solution of crystal violet in water for 30 min, the staining solution was removed, and the biofilms were washed 2x with Milli-Q water and rinsed twice with deionized water before drying the stained biofilm samples. The stained biofilms in the 5 mL tubes were photographed against a white background. The stained biofilms in the 96-well plates were incubated for 20 min in 200 µL 30% acetic acid to solubilize the crystal violet stain, which was subsequently quantified by spectrophotometry (absorbance at 590 nm).

## Bead biofilm assay

For each technical replicate (three per condition), one 6 mm natural bead (Precision Plastic Ball Company) was placed in the well of a 24-well plate, and covered with 1 mL of PAO1F in LB at an OD600 of 0.05. The plate was sealed with parafilm and shaken overnight (150 rpm, 37°C). The following day, all beads were washed by moving the bead to 1 mL sterile PBS with a forceps. Three beads were placed individually into microcentrifuge tubes with 1 mL PBS with 5 mM MgCl$_2$, sonicated for 10 min in a water bath sonicator, and titered on LB plates to determine the input. The remaining beads were transferred into fresh wells of the 24-well plate with either 1 mL LB or 1 mL LB with 10^9 pfu Clew-1 and incubated with shaking (37°C, 150 rpm) for another 2 hr. At this point, the supernatant bacteria were titered, the beads were washed 1x with PBS and placed individually into microcentrifuge tubes with 1 mL PBS with 5 mM MgCl$_2$ and sonicated and titered, as above. The bacterial titers of the supernatant and bead-associated bacteria were added to obtain the total CFU in the well and compared to the input CFU (bead titer at T0). In some experiments, the phage-treated samples were treated

with a virucide before titering (*Liu et al., 2015*). Here, 250 µL of 15%wt/vol, filter-sterilized black tea was mixed with 730 µL Milli-Q H2O and mixed with 20 µL of a freshly made, 200 mM $FeSO_4$ solution just before use (4 mM FeSO4 final). The virucide was mixed with the sample, 1:1, and incubated for 10 min at room temperature before titering the bacteria. However, this did not significantly alter the outcome of the experiment, and the data presented are a combination of all experiments, with and without virucide.

## LIVE/DEAD imaging of biofilms

Biofilms were generated in 8-well chamber slides (ibidi) by inoculating the well with 200 µL of mid-log PAO1F (diluted to an $OD_{600}$ of 0.05) and incubating the slide overnight in a humidified incubator (5% $CO_2$). The next day, the staining solution was added to each well (1 µL of a 1:10 dilution of each component in DMSO + 8 µL LB) as well as 40 µL LB (no phage control) or 40 µL LB with 10^9 pfu Clew-1. The concentrations were based on a published protocol for using LIVE/DEAD staining to test antibiotic susceptibility of biofilms (*Müsken et al., 2010*). The slides were incubated overnight and imaged using a 60x objective on a Nikon AX R laser scanning confocal microscope. The Z-stacks were analyzed and the propidium iodide positive fraction of the total biofilm volume was determined using the BiofilmQ software package (*Hartmann et al., 2021*).

## Mouse keratitis model

C57BL/6 mice were purchased from Jackson Laboratories. The mice were housed in pathogen-free conditions in microisolator cages and were treated according to institutional guidelines following approval by the University of California IACUC (Protocol# AUP-24–112). Mouse numbers are based on power calculations to estimate the number of mice needed for statistical significance (one and two-way ANOVA and Student's t-test).

Overnight cultures of *P. aeruginosa* PAO1F/pP25-GFPo were grown to log phase ($OD_{600}$ of 0.2) in LB broth, then washed and resuspended in PBS at $2.5 \times 10^7$ bacteria/ml. 7–12 weeks old C57BL/6 mice were anesthetized with ketamine/xylazine solution, the corneal epithelium was abraded with three parallel scratches using a sterile 26-gauge needle, and 2 µL of a suspension of bacteria were added topically (approximately $5 \times 10^4$ cfu per eye). After 24 hr and 48 hr, the mice were anesthetized and treated with 5*10^9 pfu CsCl purified phage Clew-1 in PBS, or PBS alone. After 72 hr the mice were euthanized, and corneas were imaged by brightfield microscopy to detect opacification, or by fluorescence microscopy to detect GFP-expressing bacteria. Fluorescent intensity images were quantified using Image J software (NIH). To determine the bacterial load, whole eyes were homogenized in PBS using a TissueLyser II (Qiagen, 30 Hz for 3 min), and homogenates were serially diluted and plated on LB agar plates for quantification of colony-forming units (CFU) by manual counting. CFU were also quantified after 2 hr to confirm the inoculum.

## Growth curves

Strains PAO1F, PAO1F Δ*fliF2*, PAO1F Δ*pslC*, and PAO1F Δ*fliF2* Δ*pslC* were grown to mid-logarithmic phase in LB, then diluted to a concentration of 10^8 cfu/mL. For growth curve measurements (OD600), three technical replicates were set up in a 96-well plate for each strain/condition. 100 µL of culture were mixed with 10 µL PBS or 10 µL with 10^8 pfu Clew-1 and incubated at 37°C in an Agilent Cytation 5 Imaging Plate Reader with a heated chamber and orbital rotation between OD600 measurements. OD600 readings were taken every 5 min.

## Culture supernatant Psl blot

Strains PAO1F Δ*fliF2* and PAO1F Δ*fliF2* Δ*pslC* were grown to mid-logarithmic phase (OD600 ~0.5), the bacteria were pelleted by centrifugation and the culture supernatant was sterilized using a 0.22 µM syringe filter. 0.5 mL supernatant samples were incubated for 1 hr at 37°C with or without 10^7 pfu Clew-1 and subsequently diluted three times at a 1:3 ratio. 2 µL of the undiluted culture supernatants and of each dilution were spotted onto a nitrocellulose filter and allowed to air-dry. The filter was then blocked with 5% non-fat milk in TBS-T (20 mM Tris Cl, 150 mM NaCl, 0.1% Tween-20) for 30 min, washed 2x with TBS-T and incubated with the primary anti-Psl antibody (diluted 1:3000) in TBS-T overnight at 4°C. The following day, the blot was washed 3x with TBS-T, then incubated with secondary antibody (horseradish peroxidase conjugated goat anti-rabbit antibody, Sigma) diluted 1:10,000 in

TBS-T for ~3 hr at room temperature. The blot was then washed 3x with TBS-T and developed using the Advansta WesternBright Sirius HRP substrate and imaged on a GE ImageQuant LAS4000 imager.

## Analysis of evolutionary relatedness

The evolutionary relationship between the Clew bacteriophage and other Bruynogheviruses was carried out using the Maximum Likelihood method and JTT matrix-based model (*Jones et al., 1992*). The tree with the highest log likelihood is shown. Initial tree(s) for the heuristic search were obtained automatically by applying Neighbor-Join and BioNJ algorithms to a matrix of pairwise distances estimated using the JTT model, and then selecting the topology with superior log likelihood value. The tree is drawn to scale, with branch lengths measured in the number of substitutions per site. This analysis involved 12 amino acid sequences. There was a total of 485 positions in the final dataset. Evolutionary analyses were conducted in MEGA11 (*Stecher et al., 2020*; *Tamura et al., 2021*). The genome comparison between Luz24 and the Clew phages was visualized using EasyFig (*Sullivan et al., 2011*).

## Acknowledgements

This work was made possible through an award from the Hypothesis fund. The authors would like to thank the Northeast Ohio Regional Sewer District, and in particular Scott Broski and Leslie Vankuren, for providing the wastewater samples from which the bacteriophage described in this study were isolated. We wish to acknowledge Sabrina Lamont and Tony DiCesare (Wozniak lab) who provided Psl preparations and αPsl rabbit polyclonal antibody for the studies described. The authors would like to thank Dr. George Dubyak for the use of his spectrophotometer/plate reader. We would like to thank Dr. Joseph Mougous for providing us with unpublished *pslC* and *pslD* complementation plasmids. We would also like to thank Dr. Mougous and Dr. Simon Dove for their support and for critical reading of the manuscript, and Dr. Matthew Parsek for his enthusiasm for the project and helpful discussions. This manuscript was supported by NIH grant R01AI169865 (to DJW), grant R01EY14362 (to EP), and grant R01 AI145069 (to EWY). The Psl-specific CAM003 antibody was obtained by G-JB from AstraZeneca (Dr. Antonio DiGiandomenico). The departmental Nikon AX R confocal microscope was supported by 1P30 DA054557 (to Alan Levine).

## Additional information

### Funding

| Funder | Grant reference number | Author |
| --- | --- | --- |
| Hypothesis Fund | | Arne Rietsch |
| National Institutes of Health | R01AI169865 | Daniel J Wozniak |
| National Institutes of Health | R01EY14362 | Eric Pearlman |
| National Institutes of Health | R01AI145069 | Edward W Yu |
| Hypothesis Fund | | Arne Rietsch |

The funders had no role in study design, data collection and interpretation, or the decision to submit the work for publication.

### Author contributions

Brenna Walton, Nikhil Vallikat, Investigation; Serena Abbodante, Data curation, Supervision, Investigation, Methodology; Michaela Ellen Marshall, Amani Alvi, Data curation, Investigation; Justyna M Dobruchowska, Daniel J Wozniak, Resources, Methodology; Larry A Gallagher, Data curation, Software, Formal analysis; Zhemin Zhang, Investigation, Visualization, Methodology; Edward W Yu, Visualization, Methodology; Geert-Jan Boons, Resources, Supervision, Funding acquisition, Methodology; Eric Pearlman, Formal analysis, Supervision, Funding acquisition, Methodology; Arne Rietsch,

Conceptualization, Resources, Data curation, Formal analysis, Supervision, Funding acquisition, Investigation, Methodology, Writing – original draft, Project administration, Writing – review and editing

**Author ORCIDs**
Michaela Ellen Marshall (ID) https://orcid.org/0000-0001-9381-780X
Eric Pearlman (ID) https://orcid.org/0000-0003-0137-7582
Arne Rietsch (ID) https://orcid.org/0000-0002-1556-7064

**Ethics**
This study was performed in strict accordance with the recommendations in the Guide for the Care and Use of Laboratory Animals of the National Institutes of Health. All of the animals were handled according to approved institutional animal care and use committee (IACUC) protocol (AUP-24-112) of the University of California, Irvine.

Reviewer #1 (Public review): https://doi.org/10.7554/eLife.102352.3.sa1
Reviewer #2 (Public review): https://doi.org/10.7554/eLife.102352.3.sa2
Author response https://doi.org/10.7554/eLife.102352.3.sa3

---

# Additional files

**Supplementary files**
Supplementary file 1. Table S1. Strains and Plasmids. Strains and Plasmids.

Supplementary file 2. Table S2. Primers used in this study. Primers.

Supplementary file 3. Table S3. TnSeq data. TnSeq data.

MDAR checklist

**Data availability**
Genomes for Clew-1, Clew-3, Clew-6, and Clew-10, as well as Ocp-2, have been deposited in Genbank [accession# PQ790658.1, PQ790659.1, PQ790660.1, PQ790661.1, and PQ790662.1].

The following datasets were generated:

| Author(s) | Year | Dataset title | Dataset URL | Database and Identifier |
|---|---|---|---|---|
| Rietsch A, Gallagher L | 2025 | Clew-1 genome | https://www.ncbi.nlm.nih.gov/nuccore/PQ790658.1 | NCBI Nucleotide, PQ790658.1 |
| Rietsch A, Gallagher L | 2025 | Clew-3 genome | https://www.ncbi.nlm.nih.gov/nuccore/PQ790659.1 | NCBI Nucleotide, PQ790659.1 |
| Rietsch A, Gallagher L | 2025 | Clew-6 genome | https://www.ncbi.nlm.nih.gov/nuccore/PQ790660.1 | NCBI Nucleotide, PQ790660.1 |
| Rietsch A, Gallagher L | 2025 | Clew-10 genome | https://www.ncbi.nlm.nih.gov/nuccore/PQ790661.1 | NCBI Nucleotide, PQ790661.1 |
| Rietsch A, Gallagher L | 2025 | Ocp-2 genome | https://www.ncbi.nlm.nih.gov/nuccore/PQ790662.1 | NCBI Nucleotide, PQ790662.1 |

The following previously published dataset was used:

| Author(s) | Year | Dataset title | Dataset URL | Database and Identifier |
|---|---|---|---|---|
| Ceyssens PJ, Demeke M, Lavigne R, Ackermann HW, Noben JP, Volckaert G, Hertveldt K | 2023 | Pseudomonas phage LUZ24, complete genome | https://www.ncbi.nlm.nih.gov/nuccore/NC_010325.1 | NCBI Nucleotide, NC_010325.1 |

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

# Appendix 1

## Appendix 1—key resources table

| Reagent type (species) or resource | Designation | Source or reference | Identifiers | Additional information |
|---|---|---|---|---|
| Strain, strain background (*P. aeruginosa*) | PAO1F | *Bleves et al., 2005* | | Wild type parent |
| Strain, strain background (Bruynoghevirus) | Clew-1 | this publication, Genbank | PQ790658.1 | Bruynoghevirus, Cleveland wastewater treatment plant |
| Strain, strain background (Pbunavirus) | Ocp-2 | this publication, Genbank | PQ790662.1 | Pbunavirus, Cleveland wastewater treatment plant |
| Recombinant DNA reagent | pEXG2 | *Rietsch et al., 2005* | | allelic exchange vector, colE1 origin, oriT, gentR, sacB |
| Recombinant DNA reagent | pPSV37 | *Lee et al., 2010* | | colE1 origin, gentR, PA origin, oriT, $p_{lacUV5}$ promoter, lacIq |
| Chemical compound, drug | Dylight594 NHS-ester | Invitrogen | 46413 | Amine-reactive abelling reagent for phage |
| Chemical compound, drug | Polystyrene beads | Precision Plastic Ball Company | | 6 mm diameter, hollow polystyrene |
| Chemical compound, drug | Filmtracer LIVE/DEAD Biofilm Viability Kit | Invitrogen | L10316 | Biofilm viability stain |
| Strain, strain background (*Mus musculus*) | C57BL/6 J | Jackson Laboratories | IMSR_JAX:000664 | Host for animal experiments |
| Software | BioflmQ | *Hartmann et al., 2021* | | MatLab program for biofilm analysis |
| Software | Prism | Graphpad | RRID:SCR_002798 | Statistical analysis software |
| Antibody | Rabbit anti-Psl antibody | *Byrd et al., 2009* | | Used for Psl immunoprecipitation |
| Antibody | Humanized anti-Psl antibody | AstraZeneca | Cam-003 RRID:AB_3111529 | Used for Psl affinity purification |
| Chemical compound, drug | Affinity purified Psl | This study | | Purification is described in the methods section |

