## [Editor Report · eLife Assessment]

This **valuable** study identifies a novel bacteriophage that can use the exopolysaccharide Psl of *Pseudomonas aeruginosa* to infect and disrupt biofilms. The work is **convincing** and suggests a novel approach to control biofilms that is relevant to researchers working on biofilms, specifically in Pseudomonas, on phage physiology and discovery, and on alternatives to controlling bacterial pathogens.

---

## [Referee Report · Reviewer #1 (Public review)]

Summary:

Walton et al. set out to isolate new phages targeting the opportunistic pathogen *Pseudomonas aeruginosa*. Using a double ∆fliF ∆pilA mutant strain, they were able to isolate 4 new phages, CLEW-1. -3, -6 and -10, that were unable to infected the parental PAO1F Wt strain. Further experiments showed that the 4 phages were only able to infect a ∆fliF strain, indicating a role of the MS-protein in the flagellum complex. Through further mutational analysis of the flagellum apparatus, the authors were able to identify the involvement of c-di-GMP in phage infection. Depletion of c-di-GMP levels by an inducible phosphodiesterase render the bacteria resistant to phage infection, while elevation of c-di-GMP through the Wsp system made the cells sensitive to infection by CLEW-1. Using TnSeq, the authors were able to not only reaffirm the involvement of c-di-GMP in phage infection but also able to identify the exopolysaccharide PSL as a downstream target for CLEW-1. C-di-GMP is a known regulator of PSL biosynthesis. The authors show that CLEW-1 binds directly to PSL on the cell surface and that deletion of the pslC gene resulted in complete phage resistance. The authors also provide evidence that the phage - PSL interaction happens during the biofilm mode of growth and that the addition of the CLEW-1 phage specifically resulted in a significant loss of biofilm biomass. Lastly, the authors set out to test if CLEW-1 could be used to resolve a biofilm infection using a mouse keratitis model. Unfortunately, while the authors noted a reduction in bacterial load assessed by GFP fluorescence, the keratitis did not resolve under the tested parameters.

Strengths:

The experiments carried out in this manuscript are thoughtful and rational, and sufficient explanation is provided for why the authors chose each specific set of experiments. The data presented strongly supports their conclusions and they give present compelling explanations for any deviation. The authors have not only developed a new technique for screening for phages targeting *P. aeruginosa*, but also highlights the importance of looking for phages during the biofilm mode of growth, as opposed to the more standard techniques involving planktonic cultures.

Weaknesses:

The authors did not include host-range testing or resistance development in this study, which would have strengthened the paper. Additionally, further characterisation of the CLEW-1 interaction with PSL at the molecular level would also have been welcomed. However, this will likely be the subject of future studies.

---

## [Referee Report · Reviewer #2 (Public review)]

This manuscript by Walton et al. suggests that they have identified a new bacteriophage that uses the exopolysaccharide Psl from *Pseudomonas aeruginosa* (PA) as a receptor. As Psl is an important component in biofilms, the authors suggest that this phage (and others similarly isolated) may be able to specifically target biofilm-growing bacteria.

Comments on revised version:

The authors have generally responded well to the reviewers' comments. This has served to improve this manuscript that has identified a new bacteriophage that uses the exopolysaccharide Psl from *Pseudomonas aeruginosa* as a receptor.

---

## [Author Response]

The following is the authors’ response to the original reviews

**Public Reviews:**

**Reviewer #1 (Public review):**
Summary:Walton et al. set out to isolate new phages targeting the opportunistic pathogen *Pseudomonas aeruginosa*. Using a double ∆fliF ∆pilA mutant strain, they were able to isolate 4 new phages, CLEW-1. -3, -6, and -10, which were unable to infect the parental PAO1F Wt strain. Further experiments showed that the 4 phages were only able to infect a ∆fliF strain, indicating a role of the MS-protein in the flagellum complex. Through further mutational analysis of the flagellum apparatus, the authors were able to identify the involvement of c-di-GMP in phage infection. Depletion of c-di-GMP levels by an inducible phosphodiesterase renders the bacteria resistant to phage infection, while elevation of c-di-GMP through the Wsp system made the cells sensitive to infection by CLEW-1. Using TnSeq, the authors were able to not only reaffirm the involvement of c-di-GMP in phage infection but also able to identify the exopolysaccharide PSL as a downstream target for CLEW-1. C-di-GMP is a known regulator of PSL biosynthesis. The authors show that CLEW-1 binds directly to PSL on the cell surface and that deletion of the pslC gene resulted in complete phage resistance. The authors also provide evidence that the phage-PSL interaction happens during the biofilm mode of growth and that the addition of the CLEW-1 phage specifically resulted in a significant loss of biofilm biomass. Lastly, the authors set out to test if CLEW-1 could be used to resolve a biofilm infection using a mouse keratitis model. Unfortunately, while the authors noted a reduction in bacterial load assessed by GFP fluorescence, the keratitis did not resolve under the tested parameters.Strengths:The experiments carried out in this manuscript are thoughtful and rational and sufficient explanation is provided for why the authors chose each specific set of experiments. The data presented strongly supports their conclusions and they give present compelling explanations for any deviation. The authors have not only developed a new technique for screening for phages targeting *P. aeruginosa*, but also highlight the importance of looking for phages during the biofilm mode of growth, as opposed to the more standard techniques involving planktonic cultures.Weaknesses:While the paper is strong, I do feel that further discussions could have gone into the decision to focus on CLEW-1 for the majority of the paper. The paper also doesn't provide any detailed information on the genetic composition of the phages. It is unclear if the phages isolated are temperate or virulent. Many temperate phages enter the lytic cycle in response to QS signalling, and while the data as it is doesn't suggest that is the case, perhaps the paper would be strengthened by further elimination of this possibility. At the very least it might be worth mentioning in the discussion section.

Thank you for your review. The genomes of all Clew phages and Ocp-2 have been uploaded [Genbank accession# PQ790658.1, PQ790659.1, PQ790660.1, PQ790661.1, and PQ790662.1]. It turns out that the Clew phage are highly related, which is highlighted by the genomic comparison in the supplementary figure S1. It therefore made sense to focus our in-depth analysis on one of the phage. We have included a supplementary figure (S1A), demonstrating that the other Clew phage also require an intact psl locus for infection, to make that logic clearer. The phage are virulent (there is apparently a bit of a debate about this with regard to Bruynogheviruses, but we have not been able to isolate lysogens). This is now mentioned in the discussion.

**Reviewer #2 (Public review):**
This manuscript by Walton et al. suggests that they have identified a new bacteriophage that uses the exopolysaccharide Psl from *Pseudomonas aeruginosa* (PA) as a receptor. As Psl is an important component in biofilms, the authors suggest that this phage (and others similarly isolated) may be able to specifically target biofilm-growing bacteria. While an interesting suggestion, the manner in which this paper is written makes it difficult to draw this conclusion. Also, some of the results do not directly follow from the data as presented and some relevant controls seem to be missing.

Thank you for your review. We would argue that the combination of demonstrating Psl-dependent binding of Clew-1 to *P. aeruginosa*, as well as demonstration of direct binding of Clew-1 to affinity-purified Psl, indicates that the phage binds directly to Psl and uses it as a receptor. In looking at the recommendations, it appears that the remark about controls refers to not using the ∆pslC mutant alone (as opposed to the ∆fliF2 ∆pslC double mutant) as a control for some of the binding experiments. However, since the ∆fliF2 mutant is more permissive for phage infection, analyzing the effect of deleting pslC in the context of the ∆fliF2 mutant background is the more stringent test.

**Recommendations for the authors:**

**Reviewer #1 (Recommendations for the authors):**
First off, I would like to congratulate the authors on this study and manuscript. It is very well executed and the writing and flow of the paper are excellent. The findings are intriguing and I believe the paper will be very well received by both the phage, Pseudomonas, and biofilm communities.

Thank you for your kind review of our work!

I have very little to critique about the paper but I have listed a few suggestions that I believe could strengthen the paper if corrected:

Comments and suggestions:

(1) The paper initially describes 4 isolated phages but no rationale is given for why they chose to continue with CLEW-1, as opposed to CLEW-3, -6, and -10. The paper would benefit from going into more detail with phage genomics and perhaps characterize the phage receptor binding to PSL.

Clew-1, -3, -6, and -10 are actually quite similar to one another. The genomes are now uploaded to Genbank [accession# PQ790658.1, PQ790659.1, PQ790660.1, and PQ790661.1]. They all require an intact Psl locus for infection, we have updated Fig. S1 to show this for the remaining Clew phage. In the end, it made sense to focus on one of these related phage and characterize it in depth.

(2) PA14 was used in some experiments but not listed in the strain table.

Thank you, this has been added in the resubmission.

(3) Would have been good to see more strains/isolates used.

We are currently characterizing the host range of Clew-1. It appears to be pretty limited, but this will likely be included in another paper that will focus on host range, not only of Clew-1, but other biofilm-tropic phage that we have isolated since then.

(4) Could purified PSL be added to make non-PSL strain (like PA14) susceptible?

We have tried adding purified Psl to a psl mutant strain, but this does not result phage sensitivity. Further characterization of the Psl receptor, is something we are currently working on, but will likely be a much bigger story than can be easily accommodated in a revised manuscript.

(5) No data on resistance development.

We have not done this as yet.

(6) Alternative biofilm models. Both in vitro and in vivo.

We agree that exploring the interaction of Clew-1 with biofilms in greater detail is a logical next step. The revised manuscript does have data on the viability of *P. aeruginosa* biofilm bacteria after Clew-1 infection using either a bead biofilm model or LIVE/DEAD staining of static biofilms. However, expanding on this further (setting up flow-cell biofilms, developing reporters to monitor phage infection, etc.) is beyond the scope of this initial report and characterization of Clew-1.

(7) There is a mistake in at least one reference. An unknown author is listed in reference 48. DA Garsin is not part of the paper. Might be worth looking into further mistakes in the reference list as I suspect this might be an issue related to the citation software.

Thank you. Yes, odd how that extra author got snuck in. This has been corrected.

(8) I don't seem to be able to locate a Genbank file or accession number. If it wasn't performed how was evolutionary relatedness data generated?

The genomes of all Clew phages and Ocp-2 have been uploaded [Genbank accession# PQ790658.1, PQ790659.1, PQ790660.1, PQ790661.1, and PQ790662.1]

(9) No genomic information about the isolated phages. Are they temperate or virulent? This would be important information as only strictly lytic phages are currently deemed appropriate for phage therapy.

These phage are virulent. We have only been able to isolate resistant bacteria from plaques, but they do not harbor the phage (as detected by PCR). This matches what other researchers have found for Bruynogheviruses.

**Reviewer #2 (Recommendations for the authors):**
Others have used different PA mutants lacking known phage receptors to pan for new phages. However, it is not totally clear how the screen here was selected for the Psl-specific phage. The authors used flagella and pili mutants and found Clew-1, -3, -6, and -10. These were all Bruynogheviruses. They also isolated a phage that uses the O antigen as a receptor. The family of this latter phage and how it is known to use this as a receptor is not described.

Phage Ocp-2 is a Pbunavirus. We added new supplementary figure S3, addressing the O-antigen receptor.

The authors focused on Clew-1, but the receptor for these other Clew phages is not presented. For Clew-1 the phage could plaque on the fliF deletion mutant but not the wild-type strain. The reason for this never appears to be addressed. The authors leap to consider the involvement of c-di-GMP, but how this relates to fliF appears to be lacking.

We have included a supplementary figure demonstrating that all the Clew phage require Psl for infection (Fig. S1A). As noted above, we have uploaded the genomic data that underpins the comparison in our supplementary figure. The phage are all closely related. It therefore made sense to focus on one of the phage for the analysis.

It is particularly unclear why this phage doesn't plaque on PAO1 as this strain does make Psl. Related to this, it actually looks like something is happening to PAO1 in Figure S4 (although what units are on the x-axis is not entirely clear).

We hypothesize that the fraction of susceptible cells in the population dictates whether the phage can make overt plaques. The supplementary figure S4 indicates that a subpopulation of the wild-type culture is susceptible and this is borne out by the fraction of wild type cells that the phage can bind to (~50%). The fliF mutation increases this frequency of susceptible cells to 80-90% (Fig. 3).

The Tnseq screen to identify receptors is clever and identifies additional phosphodiesterase genes, the deletion of which makes PAO1 susceptible. And the screen to find resistant fliF mutants identified genes involved in Psl. However, the link between the phosphodiesterase mutants and the amount of Psl produced never appears to be established. And the statement that Psl is required for infection (line 130) is never actually tested.

The link between c-di-GMP and Psl production is well-established in the literature. I think the requirement for Psl in infection is demonstrated multiple ways, including lack of plaque formation on psl mutant strains and lack of phage binding to strains that do not produce Psl, direct binding of the phage to affinity purified Psl.

Figure 2C describes using a ∆fliF2 strain but how this is different (or if it is different) from ∆fliF described in the text is never explained.

The difference in the deletions is explained in table S1, in the description for the deletion constructs used in their construction, pEXG2-∆fliF and pEXG2-∆fliF2 (∆fliF2 is smaller than ∆fliF and can be complemented completely with our complementing plasmid, pP37-fliF, which is the reason why we used the ∆fliF2 mutation going forward, rather than the ∆fliF mutation on which the phage was originally isolated).

Similarly, there is a sentence (line 138) that "Attachment of Clew-1 is Psl-dependent" but this would appear to have no context.

The relevant figure, Fig. 3, is cited in the next sentence and is the subject of the remaining paragraphs in this section of the manuscript.

For Figure 3B, why wasn't the single ∆pslC mutant visualized in this analysis? Similar questions relate to the data in Figure 4.

Analyzing the effect of the pslC deletion in the context of the ∆fliF2 mutant background, which is more permissive for phage infection, is the more stringent test.

The efficacy of Clew-1 in the mouse keratitis model is intriguing but it is unclear why the CFU/eye are so variable. The description of how the experiment was actually carried out is not clear. Was only one eye scratched or both? Were controls included with a scratch and no bacteria ({plus minus} phage)?

One eye was infected. We did not conduct a no-bacteria control (just scratching the cornea is not sufficient to cause disease). The revised manuscript has an updated animal experiment in which we carried the infection forward to 72h with two phage treatments. Following this regiment, there is a significant decrease in CFU, as well as corneal opacity (disease). Variability of the data is a fairly common feature in animal experiments. There are a number of factors, such as does the mouse blink and remove some of the inoculum shortly after deposition of the bacteria or the phage after each treatment that could explain this variability.